# Indirect Feedback Measurement of Flow in a Water Pumping Network Employing Artificial Intelligence

**DOI:** 10.3390/s21010075

**Published:** 2020-12-25

**Authors:** Thommas Kevin Sales Flores, Juan Moises Mauricio Villanueva, Heber P. Gomes, Sebastian Y. C. Catunda

**Affiliations:** 1Renewable and Alternatives Energies Center (CEAR), Electrical Engineering Department (DEE), Campus I, Federal University of Paraiba (UFPB), Joao Pessoa 58058-600, PB, Brazil; jmauricio@cear.ufpb.br; 2Technology Center (CT), Department of Civil and Environmental Engineering (DECV), Campus I, Federal University of Paraiba (UFPB), Joao Pessoa 58058-600, PB, Brazil; heberp@uol.com.br; 3Computer and Automation Engineering Department (DCA), Federal University of Rio Grande do Norte, Natal 59078-970, RN, Brazil; catundaz@gmail.com

**Keywords:** indirect reconstruction, Artificial Neural Networks, Fuzzy controller

## Abstract

Indirect measurement can be used as an alternative to obtain a desired quantity, whose physical positioning or use of a direct sensor in the plant is expensive or not possible. This procedure can been improved by means of feedback control strategies of a secondary variable, which can be measured and controlled. Its main advantage is a new form of dynamic response, with improvements in the response time of the measurement of the quantity of interest. In water pumping networks, this methodology can be employed for measuring the flow indirectly, which can be advantageous due to the high price of flow sensors and the operational complexity to install them in pipelines. In this work, we present the use of artificial intelligence techniques in the implementation of the feedback system for indirect flow measurement. Among the contributions of this new technique is the design of the pressure controller using the Fuzzy logic theory, which rules out the need for knowing the plant model, as well as the use of an artificial neural network for the construction of nonlinear models with the purpose of indirectly estimating the flow. The validation of the proposed approach was carried out through experimental tests in a water pumping system, fully automated and installed at the Laboratory of Hydraulic and Energy Efficiency in Sanitation at the Federal University of Paraiba (LENHS/UFPB). The results were compared with an electromagnetic flow sensor present in the system, obtaining a maximum relative error of 10%.

## 1. Introduction

Large systems and critical infrastructure, such as water pumping networks, oil systems, and gas pipelines, among others, require the development of a control and automation system, with the capacity to supervise and monitor the various process variables, such as flow, pressure, temperature, etc. [1]. Particularly, when it comes to water pumping systems, the cost of implementation includes purchase of the equipment (characterized by its diameter and operating pressure), assembly (acquisition of tubes, adaptation, disassembly, and necessary accessories), and electrical installation (devices protection, cables, and electrical packaging) [2]. For this kind of applications, different technologies can be used in flow measurement, such as electromagnetic, ultrasonic, and volumetric; those that employ electromagnetic principles have the highest acquisition and installation cost, depending on their diameter and consequently the volume to be measured [2,3].

An alternative method for estimating the flow can be derived from the equations that characterize the operation of a pump (pressure–flow curves) [4,5]. However, the use of this estimation method requires a calibration constant, as they are based on the constructive characteristics of the pump, which wears out during operation, varying its characteristics over time. Complementary state observer techniques can be used, which make it possible to complement the information for improving process performance, sensor and instrumentation supervision, fault detection, and filling in mass memory records associated with incomplete measurements (outliers) [6].

In these cases, where there are nonlinearities, variations in time that make mathematical modeling difficult; several studies can be found that use Artificial Neural Networks (ANN) to estimate the volume of water consumed [7], analyze the water quality in a distribution network [8,9], and apply sensor software for coagulation control in a water treatment plant [10] and virtual flow sensors [11]. Estimating ANN data associated with recording consumption in real time, they can indicate anomalies in water consumption, especially those related to water leakage. This method can also be used to estimate the “recovery” of lost data [8].

Other methods for estimating the flow consist of indirect reconstruction of the measurand from the measurement of another secondary quantity. The various methodologies of indirect estimation are based on the techniques of least squares, multiple linear regression, models based on Kalman filter, Multisensor Data Fusion (MDF), etc. [12,13]. However, these methodologies have limitations due to the numerous variables integrated in the detection and the use of approximate mathematical methods for the reconstruction of the signals, which require a high computational effort coupled with a greater possibility of failure, if the system modeling presents a low performance. To improve the performance of indirect reconstruction methods, Morawski [14] developed an indirect estimation procedure based on feedback measurement systems that implements an indirect reconstruction block, which uses a secondary quantity that can be measured and controlled to estimate a quantity of interest. In [14,15], this procedure was applied for the estimation of dissolved oxygen. It was proved and verified that the estimation dynamics can be improved by decreasing the estimation time constant. However, this methodology is based on the knowledge of the mathematical model of the plant for the controller project and the implementation of the indirect reconstruction block to perform the estimation of the quantity of interest.

Indirect estimation using artificial intelligence can be found in several other applications. Roman [16] proposed the Virtual Reference Feedback Tuning (VRFT) of a combination of two control algorithms: Active Disturbance Rejection Control (ADRC) and Fuzzy control. The main benefit of this combination is the ideal automatic adjustment in a model-free and time-saving way to find the best controller parameters. Kim [17] used an artificial neural network to linearize and group the information from three air pressure sensors with the characteristics of the contact surface size, contact force, and reference area. The accuracy and effectiveness of the tactile module were verified using real gripping experiences. With this stable grip, an ideal grip strength was estimated empirically with Fuzzy rules for a given object. Another application of virtual instrumentation is found in [18], in which the classification of vehicles on a highway is based on a piezoelectric transducer capable of performing automatic and almost instantaneous functions according to the category of vehicle traveling on the road.

This work aims to develop a feedback system for the indirect estimation of the flow of a water supply network. The applied methodology is based on the application of artificial intelligence techniques, in which a Fuzzy control system of the secondary quantity (manometric pressure) is implemented and the indirect flow is reconstructed using Artificial Neural Networks (ANN). The main contributions of this work are: (a) the design of the control system that does not require knowledge of the mathematical model of the plant and can be developed only from the knowledge of the specialist (based on rules); (b) the construction of an ANN-based indirect reconstruction block, which allows the construction of nonlinear models of the pressure inputs and the frequency of the inverter with the flow output; and (c) development of virtual sensors or soft sensors with a fast dynamic response.

Finally, the proposed procedure was tested experimentally in the water pumping system of the Laboratory of Hydraulic and Energy Efficiency in Sanitation at the Federal University of Paraiba (LENHS/UFPB), obtaining satisfactory results in the pressure control and indirect estimation procedure when compared to a Hall effect electromagnetic flow transducer.

## 2. Theoretical Background

### Indirect Measurements

Indirect measurements estimate the main quantity from an electrical signal obtained from the direct measurement of a secondary quantity that is related with the main quantity. Generally, the relationship between the secondary quantity and the main quantity is described by differential equations. As an example of application, Catunda [19] employed feedback control of the secondary quantity, resulting in a shorter response time for estimating the main quantity, and thus improved the performance of the measurement system.

Figure 1 shows the detailed diagram of the feedback measurement system, in which the measuring medium H(.) relates dynamically the main quantity, $x_{p}$, with the secondary quantity, $x_{s}$, and the acting signal, *u*. However, for the application of this methodology, two important aspects have to be taken into account: (a) the measurement method must allow the control of the secondary variable; and (b) the plant model H(.) has to provide the process variables and measurement variables.

Using a sensor, the secondary quantity $x_{s}$ is acquired and converted into an electrical signal *y* using the sensor function f(.). Then, this signal is converted to digital signal by an A/D converter providing a signal $\tilde{y}$, which is used to estimate the secondary quantity $x_{s}$, using the reconstruction function $R_{D}\left(.\right)$, which is basically the inverse function of the sensor implemented in a discrete system, given by:(1)$$x_{s}=R_{D}\left(y\right)=f^{-1}\left(y\right)$$

The Controller, D/A Converter, and Actuator blocks are responsible for controlling the secondary variable $x_{s}$ at a certain desired value (set-point). For this, the digital control algorithm c(.), which uses the measured values of the secondary quantity and the desired value, is used to generate a discrete actuation signal (*u*). This signal is converted into an analog signal (*u*) and through an actuator is applied to the plant (*H*). Finally, the main quantity is estimated in the Indirect Reconstruction block $R_{I}\left(.\right)$, using the values of the control and measurement signal of the secondary quantity, given by:(2)$$x_{p}=R_{I}\left(x_{s},u\right)$$

The actuation signal modifies the dynamics of the plant H(.), so that the response of the secondary signal $x_{s}$ follows the desired value, and therefore it will modify the dynamics in the estimation of the main quantity $x_{p}$. The convergence speed depends on the controller design.

## 3. Proposed Methodology

The flow measurement in water supply networks is essential to quantify the efficiency of these systems from a technical and economic point of view, since the acquisition of these data allows generating a historical series used for the analysis of the system’s performance. From this analysis, adjustments can be made to the operation of centrifugal pumps aiming at increasing the energy efficiency and defining planning strategies for expanding the water distribution system [15,20,21].

Figure 2 illustrates the measurement setup of the hydraulic system composed of a fully automated water pumping system, allowing the analysis of different consumption scenarios, in addition to enabling the monitoring and control of the hydraulic and electrical parameters of the system. This figure presents a legend describing the components of the experimental system. Thus, the water from the reservoir is pumped by a centrifugal pump (three-phase 220/380 V of 3 hp) through pipes and connections of Polyvinyl Chloride (PVC).

The pump forces the liquid to circulate into the system, and the liquid pressure and flow are measured. Time measurement limits of the pressure and flow transducers are 42.21 mH2O and 11.34 L/s, respectively; both are accurate to ±0.2%. The rotation speed of the centrifugal pump is controlled by means of a frequency inverter. In addition, at the system outlet, there is an automated proportional valve, which serves to emulate water demand by regulating the cross-sectional area through which the fluid passes. This experimental bench is installed in the Laboratory of Hydraulic and Energy Efficiency in Sanitation at the Federal University of Paraiba (LENHS/UFPB).

The electrical signals from the sensors are conditioned and converted into a digital signal by a data acquisition system (NI-USB 6009). Finally, these signals are sent to a desktop computer for recording measurements using the system’s supervisor. The signals that control the frequency inverter and the proportional valve are managed by the supervisory software of the same desktop computer and sent to the actuating devices through the data acquisition board, that is, converting the digital value of the command into an analog electrical signal. Figure 3 illustrates the proposed configuration of the feedback measurement system for indirect flow estimation (Q), considering as a secondary quantity the manometric pressure (P) and as the actuation variable the frequency of the inverter (*f*), which changes the rotation speed of the pump set. Then, the main modules of this proposal are discussed in detail: (a) direct reconstruction; (b) Fuzzy controller; and (c) indirect reconstruction based on ANN.

### 3.1. Pressure Measurement

The direct reconstruction block relates the measured physical variable to the electrical signal provided by the electronic circuit representing the sensor [22]. In the setup of Figure 3, this block refers to the pressure measurement using a piezoresistive transducer. This transducer is located in the discharge piping of the pumping system to monitor the controlled variable, that is, the discharge pressure. Thus, the measured pressure value corresponds to a current of 4–20 mA, which linearly corresponds to its pressure measurement range from 0 to 42.21 mH2O with accuracy of ±0.2%. This signal is converted into voltage, using a signal conditioning system operating in a dynamic range of 2–10 V.

Finally, the voltage signal is registered on a computer using a 16-bit analog/digital converter (AD), for a later reconstruction step. Therefore, the function of direct reconstruction of the secondary quantity, which represents the pressure, implements a linear inverse function of the sensor, transforming the voltage into pressure. This function can be given by:(3)$$P\left(k\right)=R_{D}\left(V\left(k\right)\right)=5.27V\left(k\right)-10.54$$
where P(k) is the pressure measurement in mH2O at the sampling instant *k* and V(k) is the voltage measured at the transducer, varying linearly in the range from 2 to 10 V (current equivalent from 4 to 20 mA).

### 3.2. Fuzzy Controller

In water supply systems, pressure control is essential for its correct functioning, since excess pressure can cause physical and financial damage, due to the rupture of the pipes [21,23]. In this way, pressure control plays a fundamental role in ensuring adequate pressure at various points of the system, in order to avoid excess pressure in the pipelines, consequently reducing water losses through leakage and avoiding the rupture of the ducts.

Thus, to solve this problem, we use the classic Proportional, Integral, and Derivative (PID) controllers, which are designed from mathematical models of the plant or tuned through tests. However, limitations are imposed regarding their use in time-varying and nonlinear systems [24]. To work around these limitations, Camboim [25] used a controller based on artificial intelligence techniques, in this case, Fuzzy logic, in which the controllers are designed from the knowledge of the system dynamics and the experience of a specialist. Experimental results in [25] show that the Fuzzy-PI controller was faster than the PID, achieving a setting time 41.26% faster, and also was more efficient with a maximum error 53% smaller than the PID.

Figure 4 illustrates the structure of the Fuzzy controller, whose objective is to generate control actions to modify the value of the dynamic pressure response at the system output. The control is of the MISO (Multiple Input Single Output) incremental type, where *P* is the control variable (pressure); P* is the reference pressure value, and represents the current error $e=P^{*}-P$; Δe is the variation of the error given by Δe=e(k)−e(k−1); Δf is the increment in the actuation signal (speed of rotation of the pump set); f(k−1) is the last value of the frequency used for actuation; and f(k) is the updated value of the frequency of the inverter, which is used as an actuation variable on the motor-pump. In this diagram, a saturator was used to prevent the motor-pump from operating outside the operating range.

The structure of the Fuzzy controller can be divided in four main blocks: fuzzyfication, rule base, inference, and defuzzyfication. In fuzzyfication, the input and output variables are represented by Fuzzy sets (error, error variation, and frequency variation), as shown in Figure 5 and Figure 6, respectively.

The limits adopted for the variable membership function of error and error variation were defined by analyzing the response of the system after starting the pump at rated speed (60 Hz), in which it was found that the maximum pressure obtained is 18 mH2O and the greatest variation in the error is 4 mH2O. The linguistic variables used for the input variables are: Negative Big (NB), Negative Medium (NM), Negative Small (NS), Zero (Z), Positive Small (PS), Positive Medium (PM), and Positive Big (PB).

On the other hand, the output variable membership function were based on the acceleration ramp of the frequency inverter used to drive the pump set, in which the maximum rotation frequency is 4 Hz. The linguistic variables of the output are: Decrease Small (DS), Decrease Medium (DM), Decrease Big (DB), Zero (Z), Increase Big (IB), Increase Medium (IM), and Increase Small (IS).

The elaboration of the rules sought to obtain a first-order response with error close to zero. These characteristics are due to the study of the system, because the presence of a high overhang and a rapid increase in the acting signal induce an excess of rotation and, consequently, a transitory pressure in the water supply network, causing the rupture of ducts and accessory devices, cavitation, and damage of the electric motor due to seasonal over current during this period.

Table 1 contains the 49 rules generated empirically by an expert, with the purpose of stabilizing the system in a permanent regime, in addition to presenting a smooth response between the exit rules according to the entry rules. For the processing of linguistic rules, the Mamdani type inference process [26] was used, and, for the defuzzyfication stage, the center of gravity method was used.

### 3.3. ANN-Based Reconstruction

According to the diagram proposed in Figure 3, the indirect flow reconstruction block ($R_{I}$) was implemented considering as inputs the estimation of the secondary quantity (pressure) that originates in the direct reconstruction block and the actuation signal of the Fuzzy controller (frequency).

For the implementation of the block $R_{I}$, one can use the relationships between the input and output variables, described by Equations (4) and (5), which define the dynamic behavior of the hydraulic network, considering the following hypotheses: unidimensional flow, linear elasticity of the tube walls, and the same correlations for the steady-state and transient pressure loss [27].
(4)$$\frac{\partial P}{\partial t}=-\frac{a^{2}}{Ag}\frac{\partial Q}{\partial x}$$
(5)$$\frac{\partial Q}{\partial t}=-gA\frac{\partial P}{\partial x}-\frac{\beta Q^{2}}{2DA}$$
where *a* is the speed of the flow, which depends on the mass density of the fluid, mode of elasticity of the pipe walls, thickness of the pipe wall, etc.; *g* is gravity; *A* is the cross-sectional area of the pipe; *Q* is the flow; *x* is the distance from the pump to the measurement point; *D* is the diameter of the pump; *P* is the head (pressure); and β is the friction factor of the fluid with the duct walls [27].

One way to calculate the friction factor is using Darcy–Weisbach equation [28]. It fundamentally describes the friction losses in tube flow as well as open channel flow. The general equation is expressed in Newtons per square meter of fluid:(6)$$\beta =\frac{\rho \beta_{f}ax}{2D}$$
where ρ is density of fluid and $\beta_{f}$ is the friction factor.

Therefore, the flow estimation can be given by solving the system of Equations (4)–(6). However, this is a nonlinear and multivariable modeling of a complex solution and with approximations that limit its generalized application. To overcome these disadvantages, this work proposes the use of Artificial Neural Networks (ANNs) to compose the indirect estimation block ($R_{I}$). The main advantages for using an ANN is the lack of mathematical models, mainly because it is a complex and multivariable problem, due to its generalization capacity, fault tolerance, self-learning, noise immunity, and adaptability [12,20,29,30,31].

In this context, for performance comparison, this work addresses two ANN topologies to compose the indirect flow estimation block ($R_{I}$), as shown in Figure 7 and Figure 8. The first ANN contains two input vectors (pump rotation speed (*f*) and pressure measurement (*P*) and an outlet (flow (*Q*)), called Multi-layer Feedforward Backpropagation ANN. The second ANN uses the same input and output vectors; however, the input vector is added by the feedback of the past value of the estimated flow (Q(k−1)), called ANN Nonlinear Autoregressive Exogenous (NARX).

The role of feedback in the ANN-NARX architecture (Figure 8) is to introduce the dynamics of the system to its learning. Thus, the indirect reconstruction block $R_{I}$ can be implemented for the indirect flow estimation, given by:(7)$$Q\left(k\right)=R_{I}\left(P\left(k\right),Q\left(k\right),Q\left(k-1\right),f\left(k\right)\right)$$

The proposed ANNs have a hidden layer with eight neurons followed by an output layer with one neuron, determined empirically in order to find the network with the least error during the training period. In addition, the activation function adopted for the hidden layer is Hyperbolic Tangent Sigmoid and that of the output layer is Linear. Finally, the training algorithm used was Levenberg–Marquardt (LM), for both of the proposed structures. The data were not normalized for a training stage.

The data used for ANN training were acquired through experimental tests of the system, where the premises were: the pump starting from rest (rotation speed equal to 0 Hz), with the proportional valve (PV) at 30° and the pressure set-point (SP) used by the Fuzzy controller ranging from 8 to 14 mH2O, rising in 2 mH2O steps and returning to 8 mH2O with the same step size. In addition, a count was set, with its initial value set to 0, as an auxiliary parameter to control the number of increments made in the reference pressure value of the controller. Steps of increasing and decreasing pressure were applied.

The experimental procedure is described in the flowchart of Figure 9, in which the values of pressure, frequency, and flow were recorded for each iteration. Following this measurement procedure, each test had an interval of 3 min, in which 1800 samples were collected at a rate of 10 samples per second, 1200 of them were for training and 600 for testing. Finally, the training had as a stopping criterion the mean square error of the cost function equal to $10^{-6}$. The results of training, testing, and validation for other input values are presented in the next section.

## 4. Experimental Results

In this section, the results of the proposed methodology are presented, evaluating the performance of the Fuzzy controller, presented in Section 3.2, for the secondary quantity (pressure), and the evaluations of the Multi-layer Feedforward Backpropagation ANN and ANN-NAXR networks, presented in Section 3.3, making comparisons regarding robustness and dynamic response. The results were collected in the water pumping system installed at the Laboratory of Hydraulic and Energy Efficiency in Sanitation at the Federal University of Paraiba (LENHS/UFPB), as shown in Figure 10.

### 4.1. Evaluation of the Fuzzy Controller

During the evaluation step of the Fuzzy controller, the pump was initially from rest, that is, with a rotation speed equal to 0 Hz and proportional valve at 30${}^{\circ }$. In addition, different pressure reference values (SP) were established: 8, 10, 12, and 14 mH2O. When the reference value equal to 8 mH2O is set, the membership function of the error and error variation input variables assumes the region Positive Large (Figure 5); as observed in Table 1, the controller’s response is an average increment. The same fact occurs when the system is stationary at 8 mH2O and the new reference value is 10 mH2O. The membership function of the error and error variation input variables assumes the region Positive Small, so the Output variable membership function is Increment Small.

The result of this experiment is illustrated in Figure 11, in which it is observed that the Fuzzy controller modified the operation point dynamic response of the secondary quantity (pressure), presenting a first-order dynamics to the variation of the desired value. In addition, the characteristics of the response during the transient and permanent regime are shown in Table 2.

### 4.2. Evaluation of the $R_{I}$ Block at the ANN Training Stage

For the implementation of the block $R_{I}$, initially the ANN Multi-layer Feedforward Backpropagation topology was considered (Figure 7), where the pump frequency and system pressure are used to indirectly estimate the flow rate. The result of the indirect flow estimation is shown in Figure 12, in which the black and gray dotted curves represent the value measured by the electromagnetic flow transducer present in the plant (Measured) and the estimated flow value by Multi-layer Feedforward Backpropagation ANN (Estimated). There is a deviation between estimated and measured values almost everywhere along the curve. This is mainly due to the lack of knowledge of the system dynamics by the ANN, that is, the lack of knowledge of the flow behavior in the previous samples.

To quantify the result obtained in Figure 12, the relative error ($E_{r}$) was calculated using Equation (Equation 6), where $v_{m}$ is the measured value and $v_{e}$ is the value estimated by ANN. Therefore, the average flow for a fixed interval during the permanent regime was considered. The results were: average of the relative error equal to 0.68%, maximum relative error equal to 9%, and standard deviation of the relative error equal to 1%.
(8)$$E_{r}=\left|\frac{v_{e}-v_{m}}{v_{m}}\right|$$

Figure 13 illustrates the comparative result between the estimated flow value and the measured value for the ANN-NARX architecture (topology shown in Figure 8). The curves between the measurements of the electromagnetic flow sensor and the estimation by indirect measurement overlap. This fact is directly linked to the introduction of the system dynamics during the learning of ANN-NARX. Thus, to quantify the results presented in Figure 13, Equation (Equation 6) was used, providing an average of the relative error equal to 8.49 × 10${}^{-5}$%, maximum relative error equal to 1.3 × 10${}^{-3}$%, and standard deviation of the relative error equal to 1.4 × 10${}^{-4}$%.

### 4.3. Evaluation of the Block $R_{I}$ at the ANN Test

From the results presented in the previous section, showing better performance during the training phase, only the ANN-NARX topology was used to compose the indirect measurement block $R_{I}$. Therefore, the system was subjected to unprecedented laboratory tests, that is, operation of the system with values that had not yet been presented during the training stage (test).

The initial conditions were as follows: the pump starting from rest, that is, with rotation speed equal to 0 Hz and proportional valve at 50${}^{\circ }$, and the pressure reference values adopted were: 8, 12, 16, 20, and 24 mH2O. First, the action of the Fuzzy controller was evaluated for different pressure reference values, where pressure measurements are illustrated in Figure 14. It is observed that the secondary quantity (pressure at the system outlet) tends to the desired reference value.

After validating the controller during the test phase, it was possible to examine the performance of the indirect flow measurement structure, as illustrated in Figure 15. Note that the estimated value curve is very close to the flow curve (values measured by the electromagnetic flow transducer), as highlighted. In a quantitative way, the relative error ($E_{r}$) was calculated using Equation (Equation 6) considering the average flow for fixed intervals during the permanent regime. The obtained results were: the average of the relative error equal to 0.13%, the maximum relative error equal to 10.43%, and the standard deviation of the relative error equal to 0.41%.

Furthermore, it is possible to observe in Figure 15 that there is a great oscillation in the flow measurement for the 60–30 s stretch in which the pressure values are 16, 20, and 24 mH2O. This is a consequence of the vibration in the ducts caused by the increasing volume of liquid that passes through the flow sensor. However, despite these fluctuations, the indirect measurement structure with ANN-NARX was able to estimate the flow value.

## 5. Conclusions

This article presents an application of artificial intelligence as an alternative to compose the control blocks and indirect flow measurement in pumping systems, based on the methodology of feedback systems. In this new approach, the design concepts of Fuzzy controllers were applied, without the knowledge of the plant model, with nonlinear characteristics, and with variations in the characteristics of the plant, due to wear and influence of disturbances during the operation of the system. Thus, ANN was also used as a computational tool for the construction of nonlinear models for the implementation of the indirect flow reconstruction block.

Pressure control in pumping systems is essential for reducing real losses and increasing energy efficiency, and it was verified that the Fuzzy controller behaved adequately, with a (maximum) steady-state error equal to 0.79%. Pressure control in pumping systems is essential for reducing real losses and increasing energy efficiency. Regarding the indirect measurement block, the use of ANN-NARX to compose this block made it possible to estimate the flow at the system output with root-mean-square error equal to 1.72%. Thus, despite the initial implementation of the automation of a pumping station presenting a relatively high cost, the methodology proposed in this work would allow the absence of flow transducers throughout the distribution system, consequently reducing the cost of acquisition and implantation of these devices.

The advantages of the proposed methodology are that it does not require the knowledge of the plant parameters to be controlled, and that the generalization capacity of ANN suits applications in complex, multivariable and time variant systems. In addition, sensor virtualization saves on the costs of deploying sensors on the physical network. However, the disadvantage is that the performance of the Fuzzy controller depends on the rules and thresholds of the functions of relevance, which are prepared by an expert. Another point is the volume of data needed for training the artificial neural network and its topology. Finally, the pumping system must be fully automated and capable of loading the algorithm into the monitoring and control system.

This methodology is in the area of sensor virtualization (soft sensors) that allows the correction and intelligent filling of invalid data (outliers), arising from flow sensors installed in the pipes, thus allowing the construction of digital twins of supply systems (Digital Twins).

In future research, other control techniques will be used, such as adaptive control, which dismisses previous knowledge of plant dynamics; the evaluation of other topologies of artificial neural networks; and the use of autoencoders for the reconstruction and correction of measurements.

## Figures and Tables

**Figure 1 sensors-21-00075-f001:**
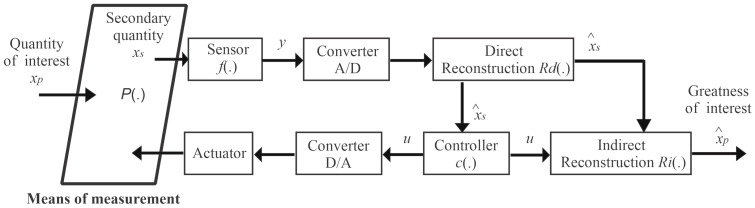
Generic diagram of a measurement feedback system.

**Figure 2 sensors-21-00075-f002:**
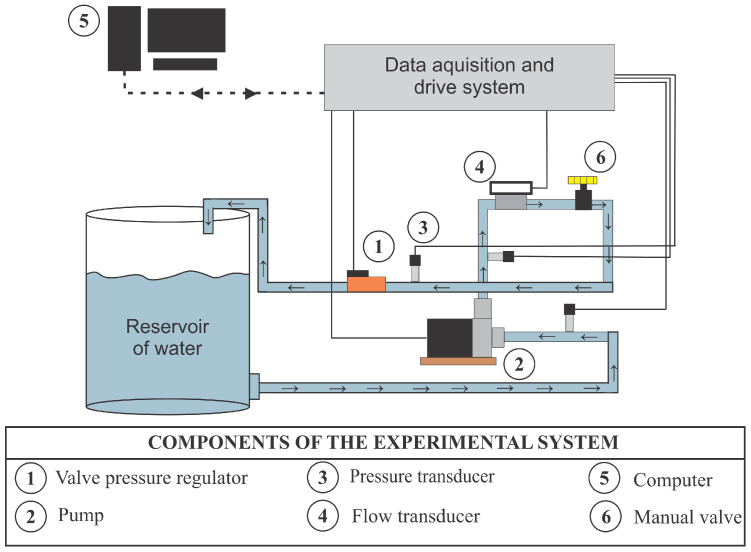
Scheme for carrying out tests and building the database.

**Figure 3 sensors-21-00075-f003:**
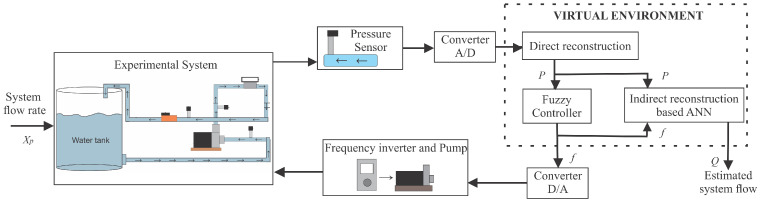
Feedback measurement system for indirect flow measurement.

**Figure 4 sensors-21-00075-f004:**
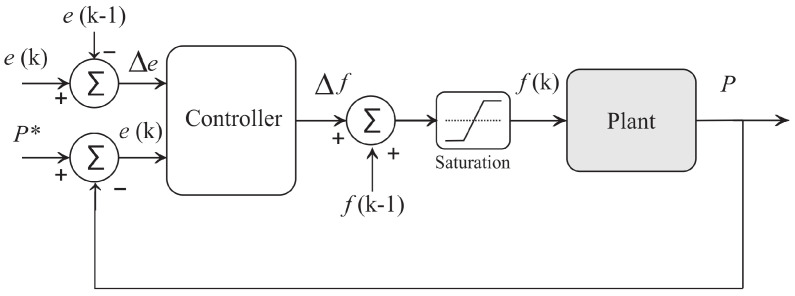
Fuzzy Pressure Control System.

**Figure 5 sensors-21-00075-f005:**
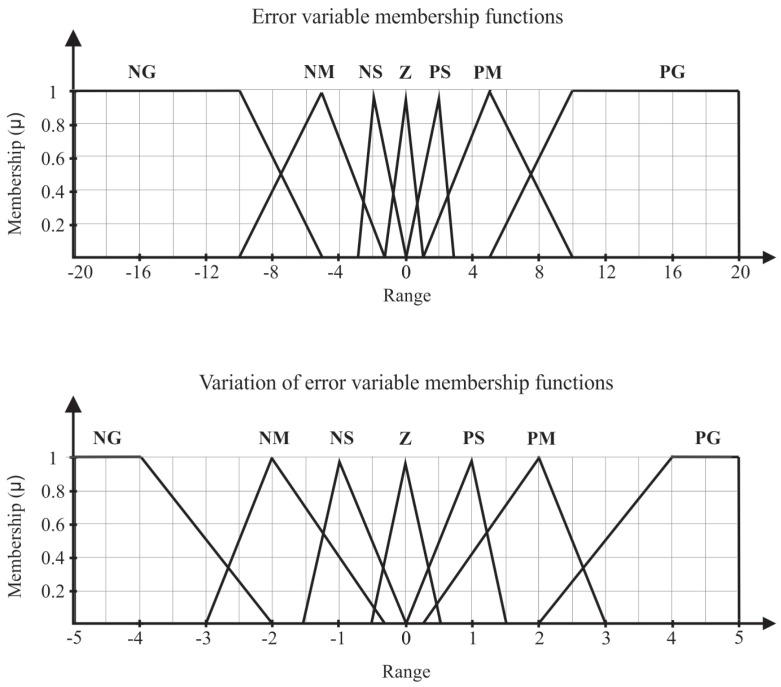
Membership function of the error and error variation input variables.

**Figure 6 sensors-21-00075-f006:**
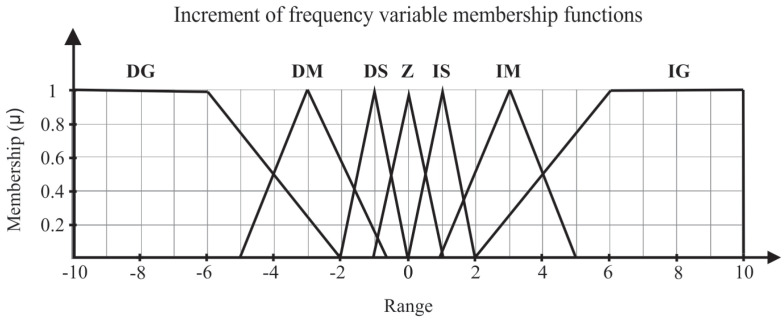
Output variable membership function.

**Figure 7 sensors-21-00075-f007:**
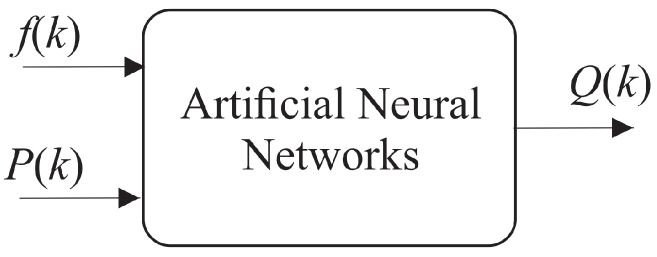
Diagram of Multi-layer Feedforward Backpropagation ANN without feedback, with two input vectors and an output.

**Figure 8 sensors-21-00075-f008:**
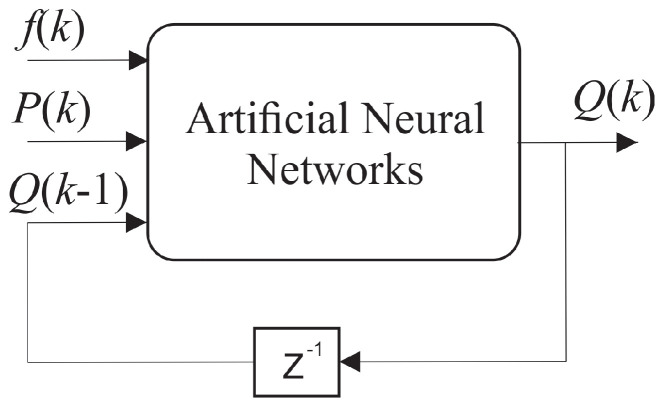
Diagram of ANN-NARX with feedback.

**Figure 9 sensors-21-00075-f009:**
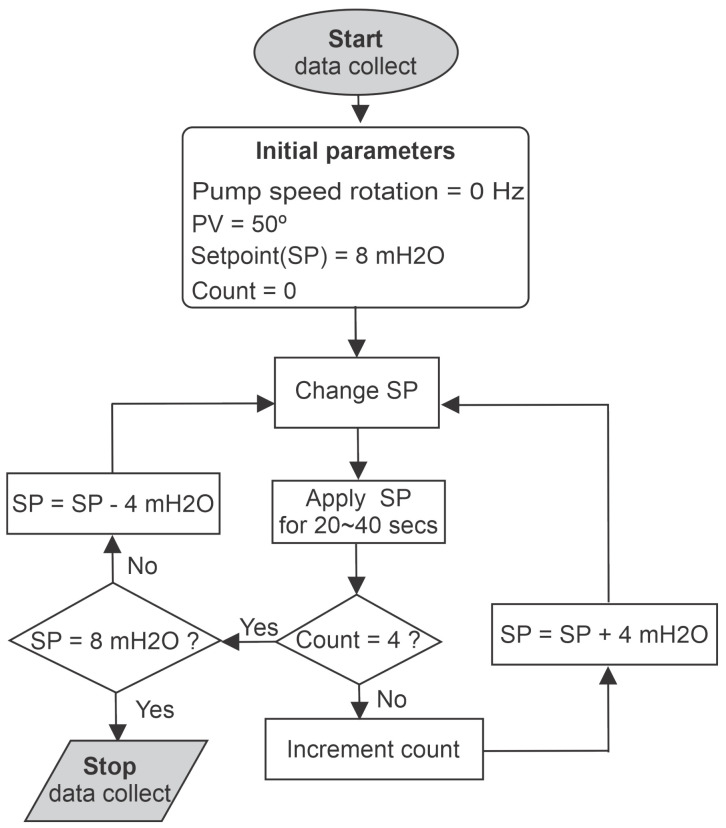
Data collection workflow for training.

**Figure 10 sensors-21-00075-f010:**
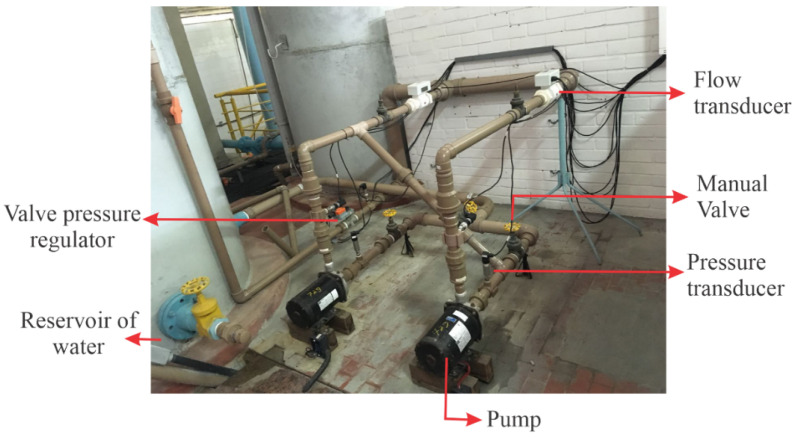
Experimental setup for data acquisition.

**Figure 11 sensors-21-00075-f011:**
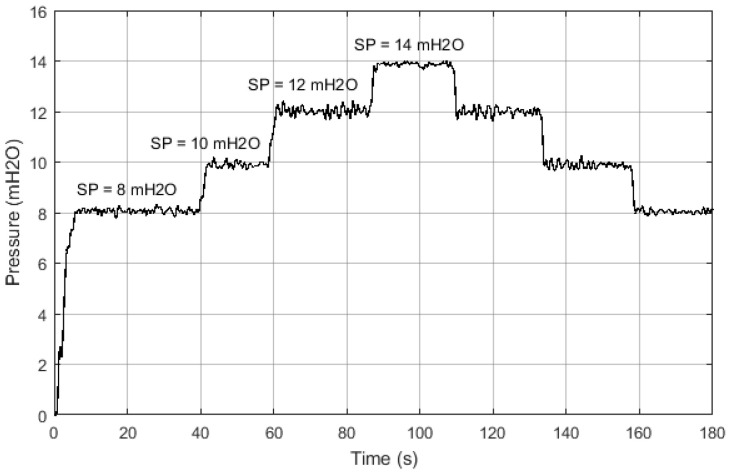
Scheme for carrying out tests and building the database.

**Figure 12 sensors-21-00075-f012:**
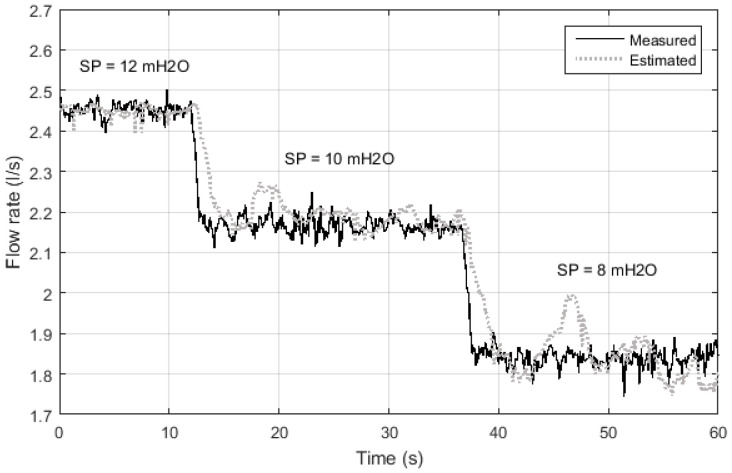
Measurement and estimation of flow during ANN testing using Multi-layer Feedforward Backpropagation.

**Figure 13 sensors-21-00075-f013:**
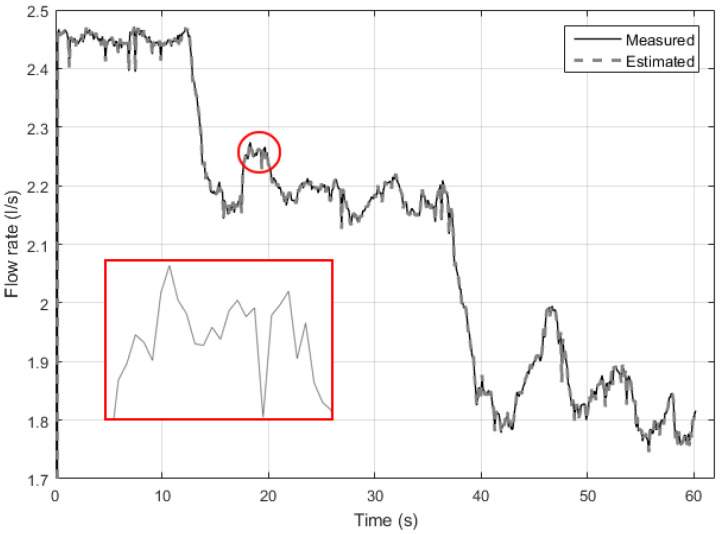
Flow measurement and estimation during ANN testing using NARX.

**Figure 14 sensors-21-00075-f014:**
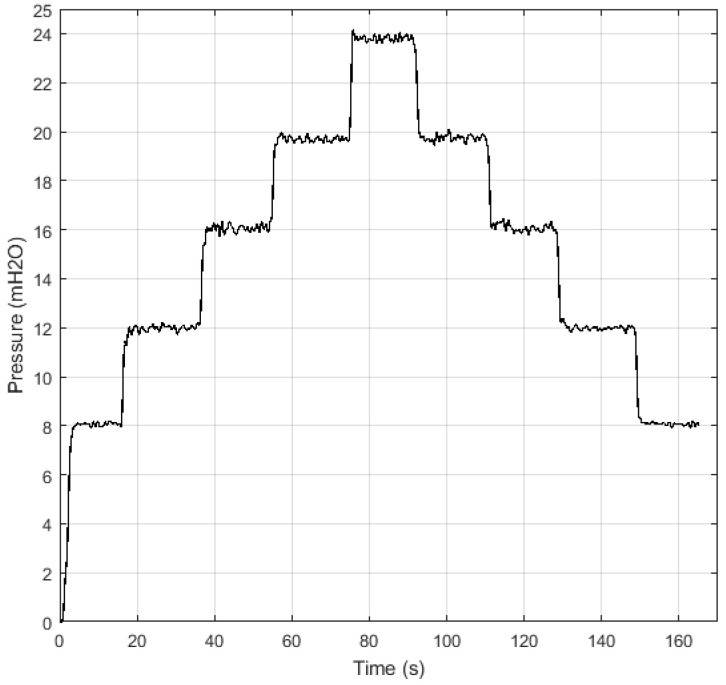
Control of the secondary quantity (Pressure) in the test phase for different set-point values: 8, 12, 16, 20, and 24 mH2O.

**Figure 15 sensors-21-00075-f015:**
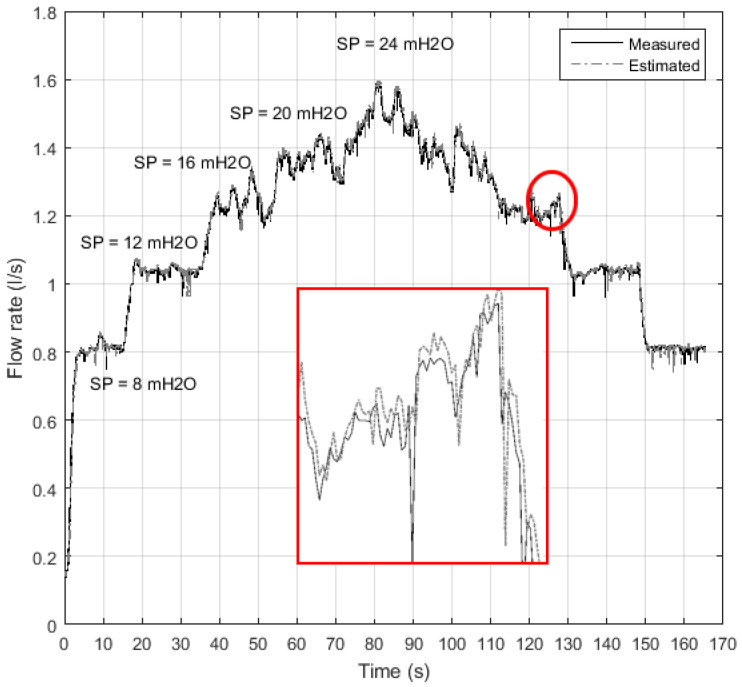
Flow measurement (sensor) and indirect flow estimation during the ANN-NARX test step.

**Table 1 sensors-21-00075-t001:** Fuzzy control rules.

	Variation of Error							
		**NB**	**NM**	**NS**	**Z**	**PS**	**PM**	**PB**
**Error**	**NB**	DS	DS	DS	DM	DM	DB	DB
	**NM**	Z	DS	DM	DM	DM	DB	DB
	**NS**	Z	Z	DS	DS	DS	DS	DM
	**Z**	IS	Z	Z	Z	Z	Z	DS
	**PS**	IS	IS	IS	Z	IS	Z	Z
	**PM**	IB	IB	IM	IM	IM	IM	IS
	**PB**	IS	IB	IB	IB	IM	IM	IM

**Table 2 sensors-21-00075-t002:** Characteristics of the transient pressure responses.

Feature	*System Response*
Rise time	1.76 s
Settling time	4.35 s
Overshoot	-
Steady-state error	0.79%

## Data Availability

Data sharing not applicable or No new data were created or analyzed in this study. Data sharing is not applicable to this article.

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
