# Peer review of "Indirect Feedback Measurement of Flow in a Water Pumping Network Employing Artificial Intelligence"

_sensors, 2020, doi:10.3390/s21010075_

Round 1
Reviewer 1 Report
sensors-1005962
Titled: Indirect feedback measurement of flow in aWater Pumping Network employing Artificial intelligence
- In line 40, what does RNA mean?
- It is necessary to indicate the accuracy of the instruments-measurements, for example the pressure.
- It is necessary to indicate the units of the variables
- How is it calculated friction factor, add Eq., if is necessary?
- In line 205, Why use an ann Multi-layer Perceptron?, If perceptron is used to determine two reponse [0,1, or -1, 1], since it uses a hard-limit or symmetric hard-limit transfer function. ANN predict the Q, value outside [0,1, or -1,1]. Remmenber you use hyperbolic tangent sigmoid transfer function.
- Indicate the accuracy of the prediction of Q with statistical tests
- The input and output data were normalized, the activation function used in multilayer were all hyperbolic tangent sigmoid?
- what was its topology found in ANN?
Author Response
We thank the reviewer comments and contributions. Annexed notes of suggested corrections.

Reviewer 2 Report
The current paper proposes the use of artificial intelligence techniques in the implementation of the feedback system for indirect flow measurement considering the design of the pressure controller using the Fuzzy logic theory, which rules out the need for knowing the plant model, as well as the use of an artificial neural network for the construction of non-linear models with the purpose of indirectly estimating the flow. The algorithm is validated through experimental results.
Comments to author:
- How the authors obtained and tuned the parameters of the fuzzy controller?
- Please add more details of how the theory from the previous sections is applied in section 4.
- The state of the art it is very poor and it should be improved with more novel references, maybe the author could add the following publications:
o Hybrid Data-Driven Fuzzy Active Disturbance Rejection Control for Tower Crane Systems, European Journal of Control, doi https://doi.org/10.1016/j.ejcon.2020.08.001, pp. 1-11, 2020.
o Multi-Agent-Based Data-Driven Distributed Adaptive Cooperative Control in Urban Traffic Signal Timing, Energies, vol. 12, no. 7, pp. 1–19, 2019.
- The authors could add a paragraph with the advantages and the disadvantages of the proposed method.
- The authors can add details about their future work.
- A section with the novelty of the current paper should be added in introduction or in the conclusion section.
Author Response

(The authors gave the same response as above.)
